# The role of oxytocin in delay of gratification and flexibility in non-social decision making

Georgia Eleni Kapetaniou[1,2]*, Matthias A Reinhard[3], Patricia Christian[1,2], Andrea Jobst[3], Philippe N Tobler[4,5], Frank Padberg[3], Alexander Soutschek[1,2]

[1]Department of Psychology, Ludwig Maximilian University Munich, Munich, Germany; [2]Graduate School for Systemic Neurosciences, Department of Biology, Ludwig Maximilian University Munich, Munich, Germany; [3]Department of Psychiatry and Psychotherapy, University Hospital, Ludwig Maximilian University Munich, Munich, Germany; [4]Zurich Center for Neuroeconomics, Department of Economics, University of Zurich, Zurich, Switzerland; [5]Neuroscience Center Zurich, University of Zurich, Swiss Federal Institute of Technology Zurich, Zurich, Switzerland

**Abstract** Oxytocin is well-known for its impact on social cognition. This specificity for the social domain, however, has been challenged by findings suggesting a domain-general allostatic function for oxytocin by promoting future-oriented and flexible behavior. In this pre-registered study, we tested the hypothesized domain-general function of oxytocin by assessing the impact of intranasal oxytocin (24 IU) on core aspects of human social (inequity aversion) and non-social decision making (delay of gratification and cognitive flexibility) in 49 healthy volunteers (within-subject design). In intertemporal choice, patience was higher under oxytocin than under placebo, although this difference was evident only when restricting the analysis to the first experimental session (between-group comparison) due to carry-over effects. Further, oxytocin increased cognitive flexibility in reversal learning as well as generosity under conditions of advantageous but not disadvantageous inequity. Our findings show that oxytocin affects both social and non-social decision making, supporting theoretical accounts of domain-general functions of oxytocin.

*For correspondence:
Georgia.Kapetaniou@psy.lmu.de

## Introduction

The neuropeptide oxytocin is well-known for its impact on social behavior, including maternal care, social recognition, or costly sharing (*Campbell, 2010*; *Lee et al., 2009*; *Macdonald and Macdonald, 2010*). Several theoretical accounts regarding the functional role of oxytocin for social cognition have been proposed. Two of the most prominent theories are the social salience hypothesis and the approach/withdrawal hypothesis. The social salience hypothesis ascribes oxytocin a crucial role for regulating attention to social cues (*Shamay-Tsoory and Abu-Akel, 2016*), whereas the social approach/withdrawal hypothesis posits that oxytocin facilitates approach-related and inhibits withdrawal-related social emotions (*Kemp and Guastella, 2011*). However, the specificity of oxytocin for the social domain is challenged by an increasing body of evidence for oxytocin effects on non-social cognition and behavior. For example, intranasal oxytocin was found to reduce craving in addiction (*Hansson et al., 2018*; *McRae-Clark et al., 2013*; *Miller et al., 2016*), reduce food intake in eating disorders (*Giel et al., 2018*), improve negative symptoms and working memory in schizophrenia (*Gibson et al., 2014*; *Michalopoulou et al., 2015*; *Pedersen et al., 2011*), and reduce avoidance of negatively valenced non-social stimuli (*Harari-Dahan and Bernstein, 2017*). From a mechanistic perspective, however, the role of oxytocin for non-social behavior remains poorly understood, given that the precise neuro-computational role of oxytocin is still a matter of controversy

(*Bethlehem et al., 2013*; *Chini et al., 2014*; *Veening and Olivier, 2013*). More recent accounts aim to reconcile the social and non-social effects of oxytocin by positing a domain-general role of oxytocin for allostasis (*Quintana and Guastella, 2020*) or for approach versus avoidance motivation (*Harari-Dahan and Bernstein, 2014*). According to the latter approach, oxytocin modulates approach-avoidance behavior by facilitating the processing of personally relevant and emotionally evocative cues. The allostatic theory of oxytocin claims that oxytocin enables maintaining stability in changing environments by facilitating the anticipation of future needs and flexible behavioral adaptations. The ability to delay gratification by resisting immediate temptation impulses is a hallmark of future-oriented behavior, whereas behavioral flexibility relies on the capacity to re-learn old, dysfunctional associations between environmental cues and outcomes (reversal learning). In fact, a role of oxytocin for delaying gratification and reversal learning could potentially explain several of the observed oxytocin effects on non-social behavior: beneficial oxytocin effects on addiction or over-eating may relate to improved delay of gratification, while deficits in re-learning of dysfunctional cue-outcome associations are a hallmark of the negative symptoms in schizophrenia (*Bowen and Neumann, 2017*; *Reddy et al., 2016*; *Waltz and Gold, 2007*). We therefore tested the impact of intranasal oxytocin on these two core aspects of non-social decision making, delay of gratification (reward impulsivity), and cognitive flexibility (reversal learning). By assessing also the impact of oxytocin on generosity (inequity aversion), we directly compare oxytocin effects on social and non-social decision making in humans and thereby bring two lines of research together that remained largely separate in the past. Thus, our study investigates the role of oxytocin beyond standard theoretical accounts focusing on its function for social cognition and behavior (*Kemp and Guastella, 2011*; *Shamay-Tsoory and Abu-Akel, 2016*).

For this purpose, we conducted a pre-registered randomized, placebo-controlled, within-subject study in which 49 healthy participants performed decision-making tasks measuring delay of gratification, reversal learning, and inequity aversion after intranasal administration of either oxytocin or placebo in two separate sessions. To explore whether the impact of oxytocin on decision making is mediated by effects on working memory capacity (WMC) (*Michalopoulou et al., 2015*), participants performed the digit span backward task as measure of WMC both before and after substance administration. Measuring WMC before substance administration allowed us to test whether the impact of oxytocin on behavior depends on baseline difference in cognitive performance, as has been observed for other pharmacological interventions particularly in the domain of reversal learning (*Cools et al., 2009*; *Kimberg et al., 1997*). We hypothesized that intranasal oxytocin (relative to placebo) increases the preference for delayed rewards in intertemporal choice, in line with previous findings from animal research and clinical studies suggesting reduced craving and impulsiveness under oxytocin (*Hansson et al., 2018*; *Hurlemann et al., 2019*; *Miller et al., 2016*). For reversal learning, we hypothesized that oxytocin improves cognitive flexibility, as suggested by animal findings (*Roberts et al., 2019*). Finally, we hypothesized that oxytocin increases advantageous inequity aversion (*Pornpattananangkul et al., 2017*).

## Results

### Oxytocin enhances delay of gratification

First, we tested the hypothesis that oxytocin improves delay of gratification (i.e., weakens the decline of the subjective value of delayed rewards with longer delays). In the intertemporal choice task, participants chose between smaller-sooner (SS; ranging from 0.5 to 4.5 euros received at the end of the experiment) and larger-later (LL; 5 euros received after a delay ranging from 1 to 180 days) reward options in 54 trials (*Figure 1A*). In a model-free analysis, we regressed choices (0 = SS option, 1 = LL option) on predictors for Substance, baseline WMC, Delay, SS reward, Order of substance administration, and all interactions. As to be expected, the analysis revealed a main effect of immediate reward, beta = −7.56, $z$ = −7.77, p < 0.001, and a significant SS reward × Delay interaction, beta = −4.57, $z$ = −4.72, p < 0.001. Moreover, the analysis suggested the presence of carry-over or task repetition effects, because order of substance administration modulated the main effect of oxytocin, beta = −2.13, $z$ = −2.68, p = 0.007, as well as the impact of oxytocin on Delay, beta = −1.97, $z$ = −2.27, p = 0.022. No other factors or interactions reached significance, all p > 0.05 (*Supplementary file 1a*). In order to control for the confounding effects of Order of

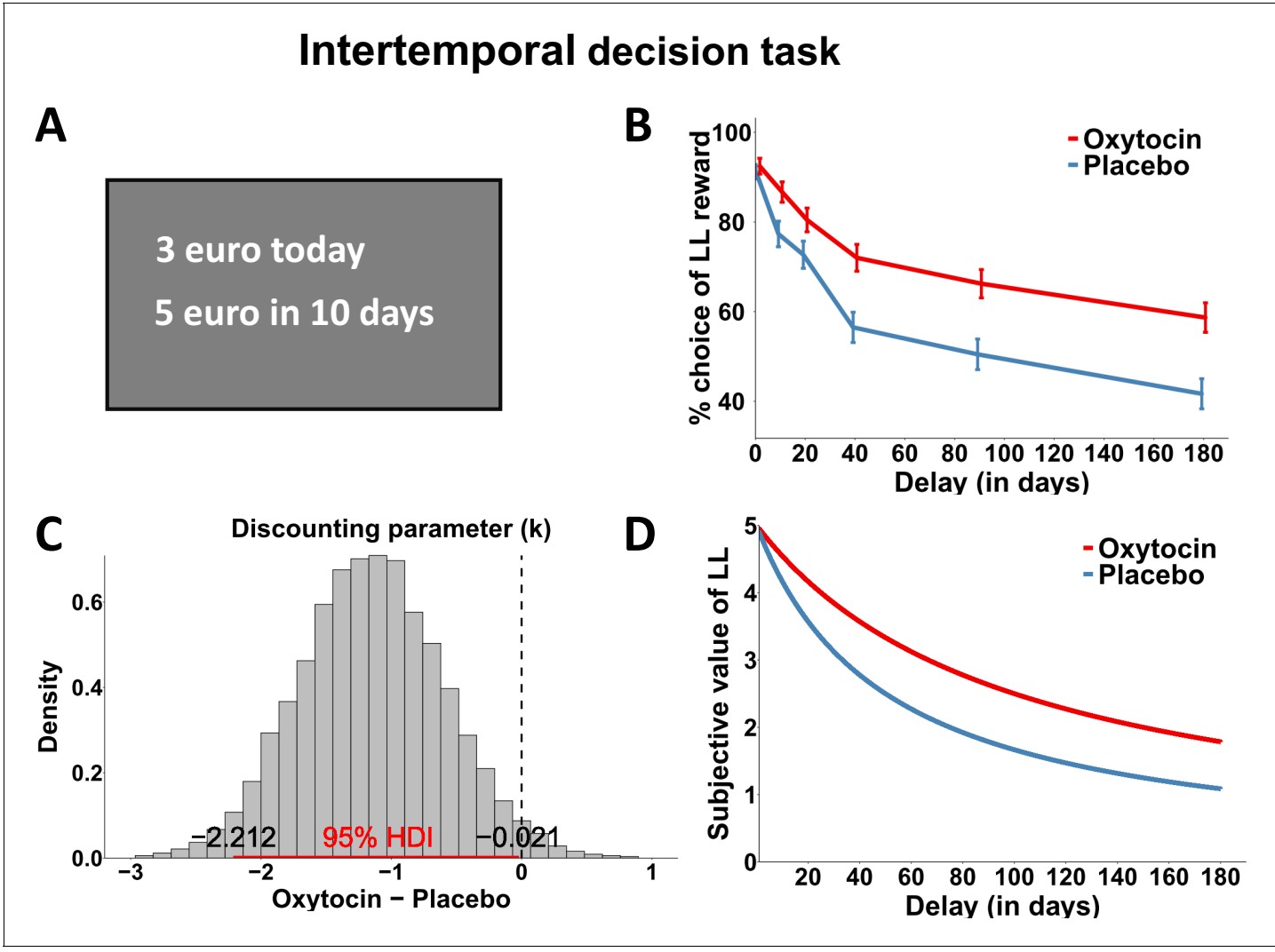

**Figure 1.** Intertemporal decision task design and results. (A) In the intertemporal choice task, participants decided between an immediate reward option (0.5–4.5 euros) and a larger-later reward option (5 euros) delivered after a delay of 1–180 days. (B) Model-free oxytocin effects on intertemporal choice. Under oxytocin, participants chose the delayed reward more frequently than under placebo (data from the first experimental session). Error bars represent standard error of the mean. (C, D) Model-based results of the intertemporal decision task. (C) Posterior distribution and 95% highest density interval (HDI) of the difference (Oxytocin – Placebo) for the discounting parameter (k). The HDI does not include 0, suggesting that the mean parameter estimates under oxytocin are lower than under placebo and that discounting of future rewards was reduced. (D) Subjective value of the delayed reward as a function of delay, based on group-level mean estimates. Participants under placebo showed overall steeper discounting of future rewards compared to oxytocin.

substance administration, we restricted our analysis on the first experimental session and re-computed the MGLM described above, leaving out predictors for Order and WMC. This procedure is recommended for crossover designs with significant carry-over effects, as the data from the first experimental session by definition are free from carry-over or task repetition effects (although at the cost of lowering the statistical power of the analyses) (*Armitage and Hills, 1982*). A sensitivity analysis indicated that our between-subject comparisons could detect effects of Cohen's $d$ = 0.81 with a power of 80% (alpha = 5%). In this analysis, we found significant main effects of Delay, beta = −4.10, $z$ = −2.28, p = 0.022, SS reward, beta = −5.99, $z$ = −6.27, p < 0.001, and the Delay × SS reward interaction, beta = −3.50, $z$ = −4.19, p < 0.001. Importantly, a significant Substance × Delay interaction, beta = 5.24, $z$ = 2.11, p = 0.034, suggests that oxytocin reduced delay discounting, supporting our hypothesis (*Figure 1B*). We found no significant interaction between Substance and SS reward, beta = −1.50, $z$ = −1.15, p = 0.248 (*Supplementary file 1b*). We note that there

was no evidence for baseline differences in WMC between oxytocin and placebo groups, $t(46)$ = 0.827, p = 0.412, such that the oxytocin effects on delay discounting are unlikely to be explained by pre-existing group differences in baseline cognitive performance.

Our model-free findings are supported by model-based analyses. Using hierarchical Bayesian modeling, we estimated the group-level hyperbolic discount parameter k (*Laibson, 1997*) in the first experimental session (due to the carry-over effects in the model-free results), separately for the oxytocin and placebo groups. To assess group differences, we computed the highest density interval (HDI) of the difference between the posterior distributions of the log-transformed group-level estimates (oxytocin minus placebo). As the 95% HDI = [−2.21; −0.02] showed no overlap with zero, we can conclude with 95% confidence that the discount rate was lower under oxytocin than under placebo (*Figure 1C, D*). In contrast, there was no evidence for group differences in the noise parameter ('inverse temperature'), HDI = [−0.20; 0.34]. Thus, the model-free and model-based findings provide converging evidence that impulsivity in intertemporal choice is reduced in the oxytocin compared to the placebo group.

## Oxytocin improves reversal learning as a function of baseline WMC

Next, we tested the hypothesis that oxytocin improves the flexible re-learning of stimulus-outcome associations. In the reversal learning task, participants were presented with two stimuli ('X' or 'O'), one of which was associated with reward (+1) and the other with loss (−1). In 120 trials, the participants were instructed to predict the outcome associated with each stimulus and use feedback to learn the correct associations, which were subject to be reversed across the task (*Figure 2A*). We regressed mean correct responses after reversal trials with predictors for Substance, Previous outcome (−1 = punishment, 1 = reward), Order, baseline WMC, and the interaction effects. While we found an interaction of Substance × Order, beta = 0.04, $t(134)$ = 2.49, p = 0.013

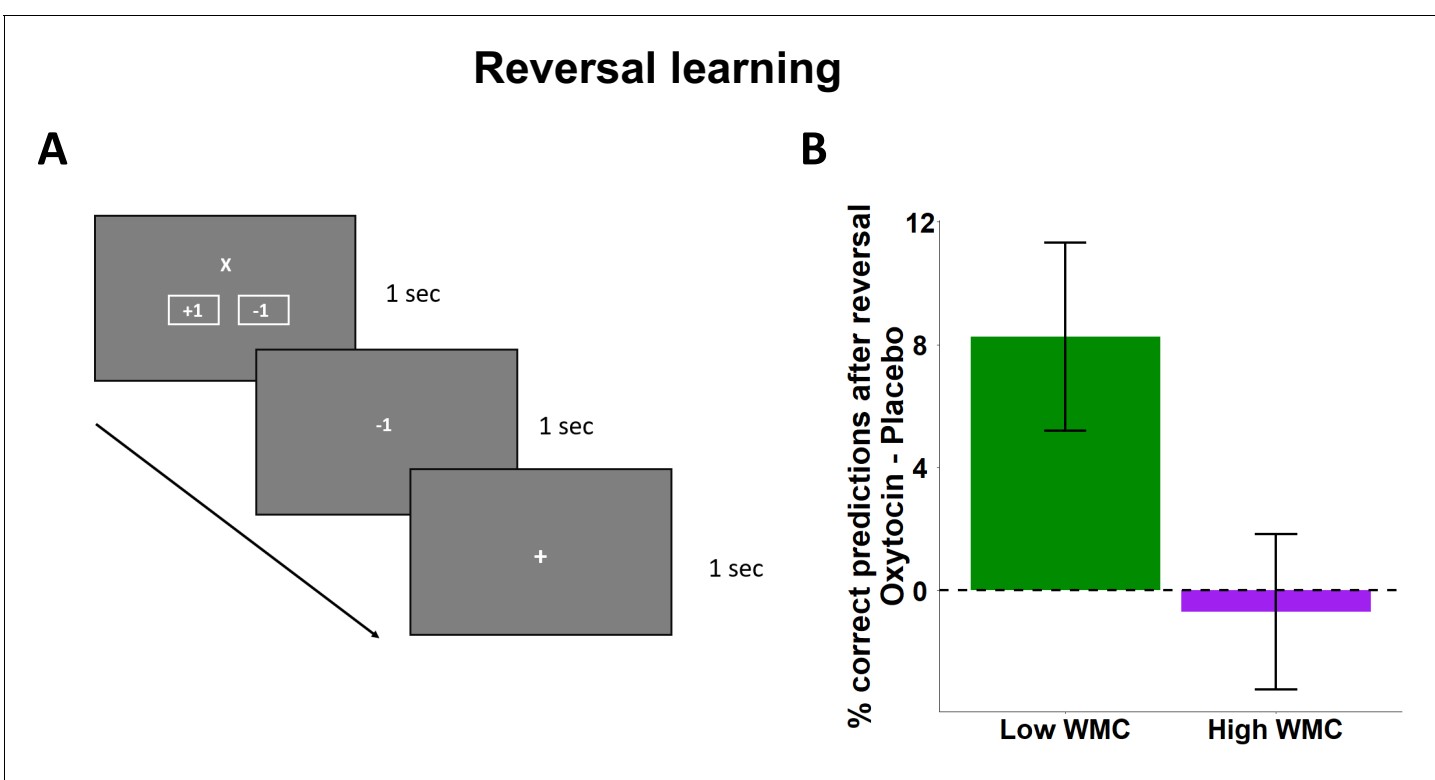

**Figure 2.** Reversal learning task design and results. (**A**) In the reversal learning task, participants were presented with one of two stimuli ('X' or 'O') and were asked to predict whether the stimulus was associated with reward (+1) or punishment (−1). Following the choice, participants viewed the outcome with which the stimulus was associated and were instructed to use this feedback to learn the correct associations. (**B**) Oxytocin increased the number of correct predictions following reversal trials relative to placebo. This effect was significant for individuals with low baseline working memory capacity. Error bars represent standard error of the mean.

(*Supplementary file 1c*), oxytocin tended to improve reversal learning relative to placebo independently of order effects, beta = 0.03, $t(134)$ = 1.90, p = 0.058, and this effect was significantly modulated by baseline WMC, beta = −0.04, $t(134)$ = −2.26, p = 0.025. No further effect was significant, all p > 0.05. To resolve this interaction effect, we performed separate analyses for the two WMC groups. While we found no significant oxytocin effects in the high WMC group, beta = −0.007, $t(68)$ = −0.27, p = 0.782, we observed significant improvement of reversal learning under oxytocin, relative to placebo, in the low WMC group, beta = 0.08, $t(66)$ = 2.69, p = 0.008 (*Figure 2B* and *Supplementary files 1d, e*). Variation in the data due to Order of substance administration in the low WMC group was accounted for by a Substance × Order interaction, beta = 0.069, $t(66)$ = 2.28, p = 0.025. These findings are in line with previous animal findings (*Roberts et al., 2019*) and suggest that oxytocin improves reversal learning as a function of baseline WMC, consistent with the baseline-dependent impacts of other pharmacological interventions (including dopamine agonists and antagonists) on reversal learning (*Cools et al., 2009*; *Kandroodi et al., 2020*; *Kimberg et al., 1997*; *Soutschek et al., 2020b*; *van der Schaaf et al., 2014*).

## Oxytocin increases advantageous but not disadvantageous inequity aversion

We further assessed whether oxytocin influences generosity under both advantageous and disadvantageous inequity. In the modified dictator game, participants chose between an equal and an unequal allocation of coins for themselves ($M_{self}$) and another randomly selected participant ($M_{other}$). The unequal allocations could be either advantageous ($M_{self} > M_{other}$) or disadvantageous ($M_{self} < M_{other}$) to the participant, allowing us to test whether oxytocin has dissociable effects on

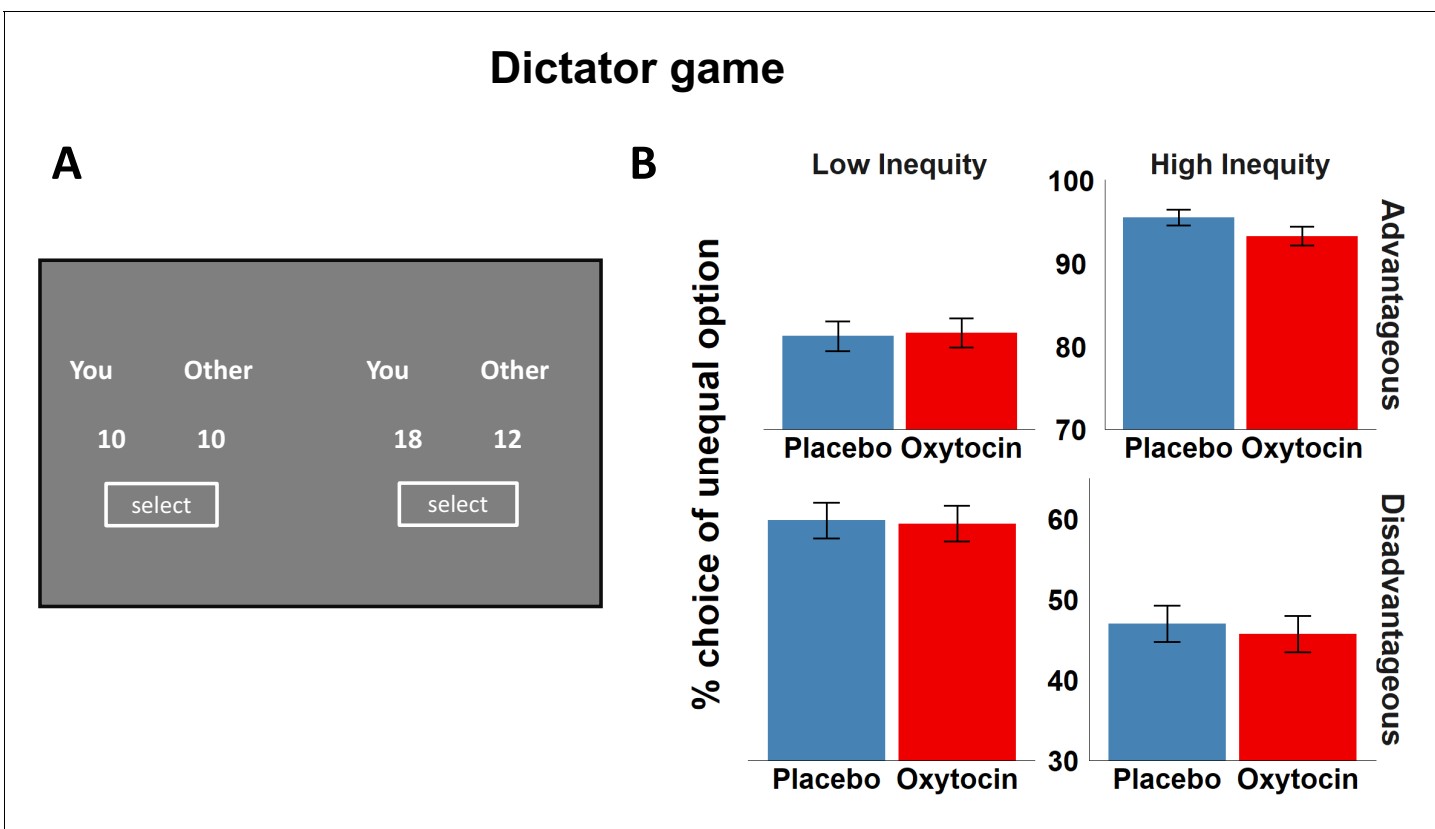

**Figure 3.** Dictator game task design and results. (A) In the modified dictator game, the participants made a choice between an equal allocation of money (You 10, Other 10) and an unequal allocation between themselves and another person. Half of the unequal allocations were advantageous for the participant (e.g., You 18, Other 12), the other half was disadvantageous (e.g., You 12, Other 18). (B) Oxytocin reduced choices of unequal reward options relative to placebo under conditions of high advantageous inequity, indicating increased aversion to being better off than others. The impact of oxytocin was significantly stronger on advantageous than on disadvantageous inequity aversion. For illustration purpose, we show oxytocin effects separately for low and high inequity trials. Error bars represent standard error of the mean.

advantageous and disadvantageous inequity aversion (*Figure 3A*). We regressed choices in the dictator game (0 = equal option, 1 = unequal option) on predictors for Substance, Order, baseline WMC, Inequity type (advantageous versus disadvantageous; the reference category was set to advantageous inequity), Inequity amount (absolute difference between $M_{self}$ and $M_{other}$), and all interaction effects, controlling for Efficiency (sum of $M_{self}$ and $M_{other}$). We observed significant effects of Inequity amount (beta = 5.32, $z$ = 3.81, p < 0.001), Efficiency (beta = 20.97, $z$ = 7.75, p < 0.001), Order $\times$ WMC (beta = 2.68, $z$ = 2.08, p = 0.036), and Order $\times$ Inequity amount $\times$ WMC interactions (beta = 1.85, $z$ = 2.13, p = 0.032) (*Supplementary file 1f*). Furthermore, participants chose the unequal option less often under conditions of disadvantageous relative to advantageous inequity (indicating stronger aversion to disadvantageous compared to advantageous inequity), main effect of Inequity type, beta = −23.09, $z$ = −6.22, p < 0.001, with this effect being even more pronounced with increasing Inequity amount, beta = −14.05, $z$ = −5.68, p < 0.001. As hypothesized, the preference for unequal over equal splits with increasing advantageous inequity was weaker under oxytocin compared with placebo, beta = −2.37, $z$ = −1.93, p = 0.026, one-tailed (as pre-registered), replicating previous findings that oxytocin increases generosity (*Pornpattananangkul et al., 2017*; *Strang et al., 2017*; *Zak et al., 2007*). We note, though, that the impact of oxytocin on advantageous inequity aversion was weaker than the effects of other task-specific experimental manipulations (e.g., Inequity amount) and would have been only marginally significant (p = 0.052) with a two-tailed test. Interestingly, oxytocin more strongly affected advantageous relative to disadvantageous inequity aversion, beta = 3.15, $z$ = 2.20, p = 0.027 (*Figure 3B*). No further effect was significant, all p > 0.05. Thus, oxytocin increases generosity, specifically in conditions of advantageous inequity, rather than inequity aversion per se.

A model-based analysis using the Fehr–Schmidt model for inequity aversion (*Fehr and Schmidt, 1999*) revealed no significant oxytocin effects, advantageous inequity aversion: 95% HDI = [−0.04; 0.04], disadvantageous inequity aversion: 95% HDI = [−0.18; 0.10]. We note, though, that the Fehr–Schmidt model may provide a poor fit of dictator game data (*Engelmann and Strobel, 2004*). Nevertheless, our model-free results replicate the finding that oxytocin increases generosity (*Pornpattananangkul et al., 2017*; *Strang et al., 2017*; *Zak et al., 2007*), extending previous findings by showing that oxytocin increases prosociality more strongly under advantageous than under disadvantageous inequity.

## No evidence for effects of oxytocin on working memory performance

The observed impact of oxytocin on social and non-social decision making raises the question as to whether these effects are mediated by a common mechanism. We explored whether working memory, as measured with the digit span backward task, might constitute such a common process, based on previous reports of oxytocin effects on working memory in schizophrenia (*Michalopoulou et al., 2015*). However, there was no evidence for significant oxytocin effects on working memory, beta = −0.003, $t$(147) = −0.12, p = 0.902 (*Supplementary file 1g*). The results further revealed significant effects of Order, beta = 0.07, $t$(100) = 2.70, p = 0.008 and Substance $\times$ Order, beta = −0.101, $t$(147) = −4.186, p < 0.001, all further effects were p > 0.05. However, even when restricting our analysis to session 1, there was no evidence for a Substance $\times$ Assessment time interaction, beta = 0.007, $t$(49) = 0.132, p = 0.89, which would suggest that oxytocin changes post-test relative to pre-test WMC. When we computed a Bayes factor indicating how strongly the data favor the alternative over the null hypothesis using the brms package (four sampling chains with 2000 iterations including 1000 warm-up iterations and normal priors with mean = 0 and sd = 1) (*Bürkner, 2017*), the Bayes factor of 0.024 indicated strong evidence in favor of the null hypothesis.

Lastly, there was no evidence for oxytocin effects on future orientation, mood, or restlessness neither on a within-subject level (across both experimental sessions; future orientation: $t$(48) = 0.78, p = 0.437; mood: $t$(48) = 0.504, p = 0.617, restlessness: $t$(48) = 0, p = 1) or a between-subject level (restricting our analysis to the first experimental session; future orientation: $t$(46) = 0.39, p = 0.697; mood: $t$(46) = 0.34, p = 0.735; restlessness: $t$(46) = 0.157, p = 0.857). It is thus unlikely that the observed oxytocin effects on decision making were driven by effects on these measures.

## Discussion

Oxytocin has been of major scientific interest for its influence on social behavior, but researchers are just beginning to explore its impact on non-social behavior (*Giel et al., 2018*; *Hansson et al., 2018*; *Miller et al., 2016*). Here, we show that intranasal oxytocin affects important components of non-social decision making, that is, delaying gratification and reversal learning.

Delay of gratification was increased under oxytocin compared with under placebo, as evidenced by both the model-based and model-free results. This result is consistent with previous evidence suggesting a link between oxytocin receptor genes and impulsiveness (*Yim et al., 2016*) as well as beneficial effects of oxytocin on impulsiveness in social anxiety disorder (although using a non-incentivized task with hypothetical rewards, hampering the validity of the measures) (*Hurlemann et al., 2019*). We speculate that oxytocin might have reduced impulsiveness via interactions with the dopaminergic system as delay of gratification has been related to dopaminergic activity (*Pine et al., 2010*; *Weber et al., 2016*), whereby blocking dopaminergic neurotransmission increases delay of gratification similar to our current findings.

Oxytocin facilitated also the re-learning of previously learned stimulus-outcome associations. This effect of oxytocin was more pronounced in individuals with low, compared with high, WMC, which mirrors the findings of dopaminergic manipulations on reversal learning (*Cools et al., 2009*; *Soutschek et al., 2020b*; *van der Schaaf et al., 2014*). Low working memory performance is associated with low dopamine baseline levels, and in fact the influence of oxytocin on learning has been hypothesized to be mediated by oxytocin-dopamine interactions (*Baracz and Cornish, 2013*). We therefore speculate that oxytocin might have improved reversal learning by strengthening valence-unspecific prediction error signals in the striatum, in analogy to previous findings for the dopamine antagonist sulpiride in a combined pharmacology-neuroimaging study (*van der Schaaf et al., 2014*). Alternatively, reversal learning also crucially depends on orbitofrontal cortex (*Schoenbaum et al., 2007*), which too is susceptible to oxytocin manipulations (*Preckel et al., 2015*). While the underlying neural mechanisms require further investigation, our results are in line with recent animal findings suggesting significant improvement in the probabilistic reversal learning task after oxytocin administration in rodents (*Roberts et al., 2019*) and provide first evidence in humans for a causal role of oxytocin for the flexible updating of cue-outcome associations. Furthermore, the result is in line with the allostatic theory (*Quintana and Guastella, 2020*) that predicts improved reversal learning under oxytocin.

Finally, we extend previous reports on the role of oxytocin for prosocial giving (*Pornpattananangkul et al., 2017*; *Strang et al., 2017*; *Zak et al., 2007*) by showing that oxytocin promotes generosity predominantly under conditions of advantageous, not under disadvantageous, inequity. Our findings therefore imply that rather than rendering behavior more altruistic or more equity-seeking in general, oxytocin strengthens the willingness to reduce self-serving inequity. Rephrased in psychological terms, our findings suggest that oxytocin reduces egocentricity when being better off than others but does not affect envy related to being worse off.

Our data raise the question as to whether common or dissociable mechanisms underlie the observed impact of oxytocin on non-social and social decision making. Oxytocin receptors are available in both prefrontal cortex and striatum (*Jurek and Neumann, 2018*), and decisions in all of the investigated domains depend on a balance between frontal and striatal networks (*Dalley et al., 2008*; *Peper et al., 2013*). It is worth noting that we found no impact of oxytocin on working memory functioning, which rather speaks against a mediating role of prefrontal activation for the observed oxytocin effects on decision making. It is thus tempting to speculate that oxytocin might have influenced decision making via oxytocin receptors expressed in striatal reward circuits, where oxytocin might mirror the effects of dopamine antagonists (*Love, 2014*). According to this view, a plausible interpretation of our findings might be that oxytocin could have increased prosocial giving and delay of gratification by lowering the subjective values of selfish and immediate rewards, respectively, whereas it may have facilitated reversal learning by enhancing neural prediction error signals. Alternatively, oxytocin might have affected decision making via interactions with the endocannabinoid system (*Pagotto et al., 2006*; *Wei et al., 2015*), which, similarly to dopamine, was linked to delay discounting and reward learning (*Boomhower and Rasmussen, 2014*; *Parsons and Hurd, 2015*).

Our findings are consistent with recent theoretical accounts ascribing oxytocin a domain-general role for both social and non-social behavior. Consistent with the predictions of the allostatic theory (*Quintana and Guastella, 2020*), oxytocin improved reversal learning and future-oriented behavior, which both may enhance an organism's survival. In this framework, aversion to advantageous inequity too may improve allostasis by strengthening social cohesion. Consistent with the neuropharmacological mechanism we proposed for our findings, optimizing behavior in changing environments relates to dopaminergic activity (*Le Heron et al., 2020*). The allostatic theory might account for the observed findings better than the approach-avoidance account (*Harari-Dahan and Bernstein, 2014*) as this account may have difficulties to explain why oxytocin reduced cost sensitivity in delay discounting instead of increasing the preference for immediate rewards (which is typically considered as approach behavior). Likewise, also the salience-based account of oxytocin would need to explain why delayed rewards and advantageous inequity are more salient than immediate rewards and disadvantageous inequity, respectively.

Alternatively, the results for delay discounting and inequity aversion might be explained by oxytocin's role for perspective taking (*Domes et al., 2007*; *Tomova et al., 2019*), given that perspective taking promotes both patient and prosocial choice (*Soutschek et al., 2016*). Perspective taking might be considered as a mechanism that enhances survival in terms of the allostatic theory, but this explanation could not explain the reversal learning results, such that the assumption that oxytocin modulates value processing appears more parsimonious. Oxytocin also reduces stress and anxiety (*Neumann and Slattery, 2016*); however, stress effects on delay discounting or reversal learning reported in the literature (*Haushofer et al., 2013*; *Joffe et al., 2019*) do not match the oxytocin effects observed in our study. On balance, oxytocin effects on the neural reward system, which on a psychological level may regulate allostatic processes, appear to be the most plausible and parsimonious explanation for the observed effects on decision making.

Our findings inform observations in clinical studies. Intranasal oxytocin has been shown to have beneficial effects on both social and non-social key symptoms of several psychiatric disorders. Regarding non-social symptoms, for example, oxytocin reduces drug craving in addiction (*Hansson et al., 2018*; *McRae-Clark et al., 2013*; *Miller et al., 2016*) and over-eating in eating disorders (*Giel et al., 2018*). Despite the evidence for such beneficial effects, a mechanistic understanding of them is limited. By providing insights into the domain-general role of oxytocin for decision making, our findings advance the field toward this direction. Given that deficits in delay of gratification contribute to the symptoms in addiction and obesity, the beneficial effects of oxytocin in these disorders might (at least partially) be caused by oxytocin-induced decreases in impulsivity. Likewise, impaired reversal learning has been associated with the negative symptoms in schizophrenia, which too may be ameliorated after oxytocin treatment (*Ota et al., 2018*). Our findings in healthy humans may thus corroborate and extend the effectiveness of oxytocin-based treatments of these disorders and suggest that it may arise at least partly through reduced delay discounting.

Some limitations are worth to be mentioned. First, we employed a systemic manipulation of oxytocin levels. As discussed above, while oxytocin effects on the dopaminergic reward system are a plausible candidate for a common neural mechanism, other mechanisms of oxytocin action with a different functional neuroanatomy (e.g., prefrontal cortex) need to be discussed as well. Furthermore, given possible carry-over effects in the delay of gratification task we followed the recommended approach of restricting our analysis to the data from the first session (*Armitage and Hills, 1982*). However, this procedure comes at a cost. First, it reduces the statistical power of our analysis due to lowering the sample size and due to relying on a between-subject instead of a within-subject comparison. Second, we cannot exclude the possibility that potential unassessed confounding variables might have driven the significant difference between the oxytocin and the placebo group. These issues could be addressed in future research by employing a parallel group design with higher statistical power or by increasing the time between the experimental sessions to lower the risk of task repetition effects. We can only speculate about the reasons for the repetition effects in the intertemporal choice task. The fact that choices were more patient in the second (mean = 73.4%) than in the first experimental session (70.6%) indicates potential anchoring effects, such that participants receiving placebo in the second session might have remembered their relatively more patient choices under oxytocin in the first session. In any case, we note that one should be cautious with ascribing oxytocin a causal role for delaying gratification due to the nature of the performed between-subject comparisons. A further limitation is that we restricted our sample to male

participants. Oxytocin may have dissociable effects on decision making and behavior in males and females (*Hoge et al., 2014*; *Kubzansky et al., 2012*), consistent with reports of gender differences in the reward system's sensitivity to the value of sharing (*Soutschek et al., 2017*). To the best of our knowledge, however, there is no evidence so far for gender differences in the neural basis of delay discounting and reversal learning, suggesting generalizability of these findings.

To conclude, our findings provide evidence for an impact of intranasal oxytocin on delay of gratification and reversal learning, demonstrating that oxytocin affects key components of both social and non-social decision making in humans. These findings contribute to the accumulating evidence challenging the specificity of oxytocin for social behavior and support recent accounts positing a domain-general role of oxytocin.

## Materials and methods

### Participants

Fifty healthy male volunteers were recruited through the participant pool of the Melessa lab at the Ludwig Maximilian University Munich, Germany. The sample size was based on a power analysis (power = 80%, alpha = 5%) assuming an effect size of Cohen's $d$ = 0.38 as observed in a previous study on the impact of oxytocin on social decision making (*Pornpattananangkul et al., 2017*). One participant dropped out after the first experimental session, resulting in a final sample of 49 participants (mean age = 23.9 years, sd = 4.14, range 18–36). The study protocol was approved by the local Ethics Committee. All participants were screened for contraindications of intranasal oxytocin and gave written informed consent before the start of the experiment. For their participation, they received 40 euros and a bonus depending on their choices. The study was pre-registered on the Open Science Framework (https://osf.io/ykvd5).

### Study design and procedures

The study followed a randomized, double-blind, placebo-controlled, crossover experimental design, spanning over two sessions timed 1 week apart. The participants were randomly allocated to two groups, one receiving oxytocin in the first and placebo in the second session, the other group receiving the substances in reversed order. Participants were assigned a subject code according to order of arrival at the lab and received the corresponding substance for that subject code in the given session. The random assignment of subject codes to drug condition was implemented by the pharmacy of the University Hospital Heidelberg and was unknown to the experimenters (double-blind design). At the beginning of the session, participants performed the digit span backward task as baseline measure of cognitive performance, followed by intranasal administration of either oxytocin or placebo. Following the standard guidelines for intranasal oxytocin administration in human participants (*Guastella et al., 2013*), the participants self-administered under supervision 24 IU (six hubs per nostril) of oxytocin (Syntocinon) or placebo, which contained the same ingredients except for the neuropeptide. Participants were unable to distinguish between oxytocin and placebo, $\chi^2(1)$ = 1.04, p = 0.307.

After a waiting period of 45 min for oxytocin to reach peak levels (*Bethlehem et al., 2013*; *Spengler et al., 2017*), participants performed a task battery including the digit span backward task, reversal learning task, dictator game, and intertemporal decision task in counterbalanced order (total task performance lasted less than 30 min). At the end, participants filled out questionnaires on demographic information, potential side effects, future orientation, mood, and restlessness. As measure for future orientation, participants rated how well they could imagine (i.e., have a clear image of) their general life situation in 10 years on a 20-point Likert scale. Mood and restlessness were measured on a 7-point Likert scale ranging from 'very bad' to 'very good' and from 'very restless' to 'very calm', respectively.

### Behavioral assessments

All tasks were programmed in zTree version 4.1.6 (*Fischbacher, 2007*).

### Intertemporal decision task

In the intertemporal choice task, participants made choices between SS and LL rewards. The amount of the immediately available SS reward varied from 0.5 to 4.5 euros in 0.5 increments (e.g., 3 euros today), whereas the LL was fixed to 5 euros and delivered after a variable delay (e.g., 5 euros in 40 days; used delays: 1, 10, 20, 40, 90, and 180 days). Crossing nine immediate amounts with six delays resulted in a total of 54 trials (*Soutschek et al., 2020a*; *Soutschek et al., 2016*). The SS and LL options were presented on the top and bottom of the screen (counterbalanced across trials), and participants made their choices by clicking with the mouse on the corresponding button (*Figure 1A*). Participants were informed in advance that one trial would be randomly selected at the end of the experiment and the chosen decision would be implemented. If a participant had chosen the SS option, the amount was paid out at the end of the experiment, whereas if he had chosen the LL option, the corresponding amount was sent to him after the corresponding delay via mail.

### Reversal learning task

We adopted a version of the reversal learning task that allows dissociating between reward and punishment reversal learning (*Cools et al., 2009*; *van der Schaaf et al., 2014*). In this task, the participants were presented with two stimuli: the letter 'X' or the letter 'O'. One of the stimuli was associated with reward, a +1 sign, and the other with a loss, a −1 sign. In a total of 120 trials, the participants were instructed to predict the outcome associated with the currently presented cue by clicking with the mouse on the corresponding button (*Figure 2A*). After the selection, the correct association appeared on the screen and the participants were instructed to use this feedback to learn the correct associations. They were instructed, however, that these associations may change within the task and they should again use the feedback to learn the new associations as quickly as possible. After such reversals, participants faced an unexpected punishment after selecting a stimulus previously associated with reward or unexpected reward after selecting a stimulus previously associated with punishment. Accuracy on the trials following reversals is thought to reflect the ability to update stimulus-outcome associations after unexpected outcomes (rewards or punishments).

### Dictator game

In the modified dictator game (*Gao et al., 2018*), participants chose between allocations of coins for themselves and a randomly selected anonymous participant in the room ('other') in order to assess inequity aversion. Each trial included two options, one with an equal allocation of coins between the participant and the other ($M_{self} = M_{other}$, e.g., 'You 10 and Other 10'), the other option with an unequal allocation (*Figure 3A*). In half of the trials, the unequal allocation was advantageous for the participant (e.g., 'You 18 and Other 12') and in the other half disadvantageous (e.g., 'You 12 and Other 18'), allowing to dissociate between advantageous and disadvantageous inequity aversion. Generosity is indicated either by increased advantageous inequity aversion or lower disadvantageous inequity aversion. In a total of 42 trials, participants were presented with different combinations of $M_{self}$ and $M_{other}$, with $M_{self}$ and $M_{other}$ ranging from 2 to 30 coins. The position of the two options on the screen was counterbalanced, and participants made their choice via mouse-click on the corresponding button. Experimental coins were translated to money at an exchange rate of 4 coins to 1 euro, and one randomly selected trial was paid out at the end of the experiment. Thus, each participant received the payoff selected for himself in the given trial and the amount another participant had selected for the other person.

### Digit span backward task

Participants performed the digit span backward task before and after substance intake in both sessions. In this task, participants were presented with a series of numbers displayed separately on the screen and were asked to write the numbers in the reverse order. This task represents a widely used measure of WMC as it requires both the maintenance and active manipulation of items in working memory. The difficulty increased gradually from 3 to 10 digits. This task allowed to assess whether the effects of oxytocin on decision making are mediated by potential oxytocin effects on WMC, as reported in schizophrenia (*Michalopoulou et al., 2015*, but see *Bradley et al., 2019*), and whether the strength of oxytocin effects on the decision making task varies as a function of baseline cognitive

performance, similar to other pharmacological manipulations of value-based choice (*Cools et al., 2008*; *Kimberg et al., 1997*; *Soutschek et al., 2020b*).

## Data analysis

### Model-free analyses

Statistical analyses were performed with R version 3.6.0 (*R Development Core Team, 2019*). The alpha threshold was set to 5% two-tailed for all analyses except for the dictator game where we used a one-tailed test (as pre-registered) to replicate previous findings that oxytocin increases generosity (*Pornpattananangkul et al., 2017*). Given these previous findings, we would consider both a too-weak effect in the expected direction and an effect in the unexpected direction as failed replication of previous reports that oxytocin increases advantageous inequity aversion. All data supporting the findings of this study are available on the Open Science Framework (https://osf.io/yg7ah/files/).

To assess whether oxytocin effects depend on baseline cognitive performance levels, we used WMC as a proxy of baseline cognitive performance. To calculate individual WMC scores, we summed the correct responses over the two pre-test assessments in the digit span backward task. This variable was normally distributed with mean = 8.69, median = 9, range = 4–14, and sd = 2.45. We used a binary WMC variable where participants with performance above and below the median were categorized as high and low WMC group, respectively. We used WMC as binary rather than continuous predictor to reduce the impact of outliers in WMC performance on statistical results (*Kimberg et al., 1997*; *Soutschek et al., 2020b*).

For all model-free analyses, we used the lme4 package in R for mixed generalized linear models (MGLMs) (*Bates et al., 2015*). All MGLMs included dummy-coded predictors for Substance (0 = placebo, 1 = oxytocin), WMC (−1 = low, 1 = high), and Order of substance administration (−1 = placebo in session 1 and oxytocin in session 2, 1 = oxytocin in session 1 and placebo in session 2). Substance × WMC interactions modeled potential baseline-dependent effects of oxytocin, whereas Substance × Order interactions allowed to statistically detect and control for potential order or carry-over effects of drug administration. All within-subject fixed effect predictors were also modeled as random slopes in addition to participant-specific random intercepts. Detailed results tables for all models are reported in *Supplementary file 1*.

### Model-based analyses

For the intertemporal decision and the dictator game tasks, we also conducted model-based analyses. The benefit of model-based analyses is that they allow assessing how oxytocin affects latent psychological processes (e.g., that decision makers integrate rewards and delays to hyperbolically discounted subjective reward values) (*Forstmann et al., 2011*; *Konovalov et al., 2018*; *Soutschek et al., 2020a*). Model-free analyses, in contrast, assess the impact of experimental manipulations (e.g., reward magnitude and delay) on choice behavior without making any assumption regarding the underlying psychological processes.

We performed model-based analyses for the intertemporal decision task and the dictator game data using hierarchical Bayesian modeling with the hBayesDM package version 1.0.2 (*Ahn et al., 2017*) and Stan version 2.19.1 (*Carpenter et al., 2017*). While individual maximum likelihood parameter estimation often results in noisy parameter estimates, hierarchical Bayesian modeling estimates group-level hyperparameters in addition to individual estimates. This leads to more stable and reliable parameter estimates as it allows individual parameters to be informed by the group-level hyperparameters (*Ahn et al., 2017*). For parameter estimation, we used four Markov Chain Monte Carlo sampling chains with 5000 iterations (including 1000 warm-up iterations).

As parameter estimates from hierarchical Bayesian modeling violate the independence assumption of frequentist inference statistics, we assessed group differences (oxytocin versus placebo) with the HDI of the posterior distribution of the difference between group-level hyperparameters. The HDI corresponds to the range of the difference between the group-level posterior distributions that spans 95% of the distribution, thus the range that entails the difference in group-level hyperparameter with 95% probability. If the HDI does not overlap with zero, the parameter estimates are considered to differ between the groups (*Ahn et al., 2017*; *Ahn et al., 2014*; *Kruschke, 2013*). Note that this procedure is not equivalent to frequentist null hypothesis testing, but it can be interpreted in a similar way.

For the *intertemporal choice* data, we computed hyperbolic discounting functions indicating the discounted value of delayed rewards as a function of delay. In hyperbolic discounting, the subjective value of reward x delivered after delay D is given by the following function:

$$SV(x,D) = \frac{x}{1+kD}$$

where k corresponds to the individual discounting rate. Greater k indicates greater discounting of future rewards (*Laibson, 1997*). To estimate the individual and group-level parameters, we used the default options of the hBayesDM toolbox, that is, normal prior distributions with mean 0 and standard deviation of 1, parameter bounds for k between 0 and 1 and starting value at 0.1. For the inverse temperature parameter, the bounds were set to 0 (lower) and 5 (upper) and starting value at 1.

For the *dictator game* data, we used the Fehr–Schmidt model for inequity aversion (*Fehr and Schmidt, 1999*). This is a widely used model of social preferences that allows dissociating between advantageous and disadvantageous inequity aversion. According to the model, the subjective value of an option depends on both one's own payoff and the payoff of the other according to the following formula:

$$SV\left(M_{self}, M_{other}\right) = M_{self} - \alpha \max\{M_{other} - M_{self},\ 0\} - \beta \max\{M_{self} - M_{other},\ 0\}, \quad M_{self} \neq M_{other}$$

where $M_{self}$ is the decision maker's own payoff and $M_{other}$ is the other's payoff in a given trial. Parameters $\alpha$ and $\beta$ reflect the weight given to disadvantageous and advantageous inequity, respectively. For the parameter estimation, we used normal prior distributions with mean 0 and standard deviation of 1 for all parameters. All parameters were unbounded with random starting values.

## Acknowledgements

We kindly thank the Munich Experimental Laboratory for Economic and Social Sciences (MELESSA) for support with recruitment and data collection.

## Additional information

### Competing interests

Matthias A Reinhard: MAR is supported by the FöFoLe program (grant #996) and FöFoLePLUS program (grant #003, MCSP) of the Faculty of Medicine of the Ludwig Maximilian University, Munich. Frank Padberg: Frank Padberg is a member of the European Scientific Advisory Board of Brainsway Inc., Jerusalem, Israel, and has received speaker's honoraria from Mag&More GmbH and the neuroCare Group. His lab has received support with equipment from neuroConn GmbH, Ilmenau, Germany, and Mag&More GmbH and Brainsway Inc., Jerusalem, Israel. Alexander Soutschek: AS received an Emmy Noetherfellowship (SO 1636/2-1) from the German Research Foundation. The other authors declare that no competing interests exist.

### Funding

| Funder | Grant reference number | Author |
| --- | --- | --- |
| Deutsche Forschungsgemeinschaft | Emmy Noether fellowship (SO 1636/2-1) | Alexander Soutschek |
| Ludwig-Maximilians-Universität München | FöFoLe program (grant #996) and FöFoLe PLUS program (grant #003, MCSP) | Matthias A Reinhard |
| Schweizerischer Nationalfonds zur Förderung der Wissenschaftlichen Forschung | Grants 100014_165884 and 100019_176016 | Philippe N Tobler |

The funders had no role in study design, data collection and interpretation, or the decision to submit the work for publication.

## Author contributions
Georgia Eleni Kapetaniou, Conceptualization, Data curation, Software, Formal analysis, Investigation, Visualization, Methodology, Writing - original draft, Project administration; Matthias A Reinhard, Conceptualization, Investigation, Methodology; Patricia Christian, Investigation, Methodology; Andrea Jobst, Philippe N Tobler, Frank Padberg, Conceptualization, Methodology; Alexander Soutschek, Conceptualization, Resources, Data curation, Software, Formal analysis, Supervision, Funding acquisition, Investigation, Visualization, Methodology, Writing - original draft, Project administration

## Author ORCIDs
Georgia Eleni Kapetaniou (iD) https://orcid.org/0000-0002-0020-1158
Alexander Soutschek (iD) https://orcid.org/0000-0001-8438-7721

## Ethics
Human subjects: All participants gave written informed consent before the start of their participation in the study. The study protocol was approved by the Ethics Committee of the Department of Psychology, LMU Munich.

## Decision letter and Author response
Decision letter https://doi.org/10.7554/eLife.61844.sa1
Author response https://doi.org/10.7554/eLife.61844.sa2

# Additional files

## Supplementary files
- Supplementary file 1. Full regression models output.
- Transparent reporting form

## Data availability
All data generated and analyzed during this study are included in the manuscript. Source data files and code for the statistical analyses and graphs are available online, on the Open Science Framework (https://osf.io/yg7ah/).

The following dataset was generated:

| Author(s) | Year | Dataset title | Dataset URL | Database and Identifier |
|-----------|------|---------------|-------------|-------------------------|
| Kapetaniou GE, Reinhard MA, Christian P, Jobst A, Tobler PN, Padberg F, Soutschek A | 2020 | Oxytocin improves delay of gratification and cognitive flexibility in non-social decision making | https://osf.io/yg7ah/files/ | Open Science Framework, yg7ah |

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
