## [Decision Letter]

**Acceptance summary:**

The editors and reviewers agreed that this is an interesting and convincing demonstration that intranasal administration of oxytocin can impact non-social processing in human participants. Specifically, oxytocin was found to increase patience during delay discounting and improve reversal learning, although the latter finding was observed only in those participants with relatively poorer working memory. Oxytocin was also found to affect social processing, as expected, increasing generosity on an inequality aversion task. The within-subjects design provides good experimental sensitivity, although repetition effects on the delay aversion task meant that a less sensitive between-subjects analysis was required; nonetheless a substantial effect was observed. The study and directional hypotheses were pre-registered, which increases confidence in the findings, and the authors have made their data and analysis scripts publicly available.

**Decision letter after peer review:**

Thank you for submitting your article "Oxytocin improves delay of gratification and cognitive flexibility in non-social decision making" for consideration by *eLife*. Your article has been reviewed by 3 peer reviewers, and the evaluation has been overseen by a Reviewing Editor and Christian Büchel as the Senior Editor. The following individual involved in review of your submission has agreed to reveal their identity: Franziska Plessow (Reviewer #2).

The reviewers have discussed the reviews with one another and the Reviewing Editor has drafted this decision to help you prepare a revised submission.

As the editors have judged that your manuscript is of interest, but as described below that additional analyses are required before it is published, we would like to draw your attention to changes in our revision policy that we have made in response to COVID-19 (https://elifesciences.org/articles/57162). First, because many researchers have temporarily lost access to the labs, we will give authors as much time as they need to submit revised manuscripts. We are also offering, if you choose, to post the manuscript to bioRxiv (if it is not already there) along with this decision letter and a formal designation that the manuscript is "in revision at *eLife*". Please let us know if you would like to pursue this option. (If your work is more suitable for medRxiv, you will need to post the preprint yourself, as the mechanisms for us to do so are still in development.)

Summary:

The reviewers agreed that this was an interesting study relevant to emerging research pointing to a non-social role for oxytocin. They felt that the topic is timely and of broad interdisciplinary interest, that the writing is clear, and that the intertemporal decision task results are striking. The reviewers also commended the authors on pre-registering their hypotheses and recruitment plans, and for posting their data and analysis scripts on OSF. However they raised a number of queries and suggestions, summarised below.

Essential revisions:

1. While the authors provide a rationale for focusing on session 1 only for the delay discounting analysis (due to the observed asymmetric practice effects), this step transforms the original within-subjects design into a between-subjects approach. This is a problematic feature of the design that deserves more attention, since it poses potential challenges in terms of data interpretation. The lack of group differences in working memory capacity, future orientation, restlessness, or mood is reassuring – however, the power for these contrasts is low and the authors cannot exclude the possibility that umeasured confounders could contribute to pre-existing group differences in delay discounting. This needs to be explicitly acknowledged in the Discussion section, and a better justification should be provided for discarding half the data. It may be that only leaving one week in between sessions was too short, and this limitation should also be mentioned.

2. It would be instructive to conduct some sensitivity analyses. For example the authors could tatistically control for order effects when analysing all the data in the planned within-subjects analysis to see whether the results converge. It would also be useful to report the required effect size for reasonable power (e.g. 80%) for the between-subjects analyses.

3. The causal interpretation implied in the title and text, for example, "our results suggest a causal link between oxytocin and delay of gratification" (p. 11, paragraph 3), are difficult to justify based on the data reported (between-groups analysis and lack of data that prove comparability between groups at a baseline). Any casual reader only glancing at the title, abstract, and Discussion section only would assume that oxytocin increased delay of gratification within subjects, as the between-subjects analysis is not made clear. The authors should revise the title, abstract, and Discussion section to to clearly distinguish between the findings that allow for a clear causal interpretation (e.g., the impact of oxytocin on reversal learning when controlled for baseline WMC) versus those that might be confounded by pre-existing differences (e.g., the session 1 temporal discounting differences).

4. Is it correct that only WMC was assessed prior to and following drug administration? Were future orientation, restlessness, and mood were also measured before drug administration? If no, it is not appropriate to report these variables as reflecting baseline characteristics [p. 5, paragraph 2]. If yes, effects of oxytocin differences on mood, restlessness and future orientatiion should be reported (whether significant or otherwise). While there is little evidence that a single intranasal oxytocin dose has an effect on mood restlessness, it would be important to rule out such effects.

5. The justification for using a one-tailed p-value in the analysis of inequity preference is weak: the p value of.026 would not be significant two-tailed, and the fact that previous studies have pointed in the same direction is not a sufficiently strong justification. The main effects of inequity amount and inequity type on the outcome of the dictator game seem many times stronger than the weak and non-significant modulating effect of oxytocin.

6. The analyses contain a large number of main effects and interactions. For full transparency it is important to report these for example by including tables showing the complete models (this could be included in a supplement). It should be explicitly stated whether any effects other than those reported reached significance, for example using wording such as "no other factors or interactions reached significance, all p>.05."

7. There needs to be some discussion of the fact that oxytocin induced a practice effect in delay discounting but not the other tasks. For example, reversal learning often results in practice effects, yet, in this study these did not appear to interact with oxytocin. Do the authors consider this to be meaningful, and what could account for such differences across tasks? How could this set of results inform future research?

8. The Discussion section would be strengthened if there was a greater focus on discussing the results that were observed. There is quite a lot of speculation (e.g., "by strengthening valence-unspecific prediction error signals in the striatum…") that is rarely labelled as such. The speculation needs to be toned down considerably, and where necessary it should be flagged with language such as "We speculate that…".

---

## [Author Response]

Essential revisions:1. While the authors provide a rationale for focusing on session 1 only for the delay discounting analysis (due to the observed asymmetric practice effects), this step transforms the original within-subjects design into a between-subjects approach. This is a problematic feature of the design that deserves more attention, since it poses potential challenges in terms of data interpretation. The lack of group differences in working memory capacity, future orientation, restlessness, or mood is reassuring – however, the power for these contrasts is low and the authors cannot exclude the possibility that umeasured confounders could contribute to pre-existing group differences in delay discounting. This needs to be explicitly acknowledged in the Discussion section, and a better justification should be provided for discarding half the data. It may be that only leaving one week in between sessions was too short, and this limitation should also be mentioned.

We thank the reviewers for this comment. In the revised manuscript, we added a more detailed justification for focusing on session 1 only. According to the recommendations for analyzing crossover designs (Armitage and Hills, 1982), it is first important that authors statistically assess and control for potential carry-over effects in addition to the treatment effects per se, which we do by modelling the Substance × Order interactions in all statistical models. If an analysis yields no significant treatment effects but only significant carry-over effects (in our case, Substance × Order interactions), it is recommended to analyze only the data from the first experimental sessions, as these by definition are free from drug order effects, even though this comes at the cost of reduced statistical power (Armitage and Hills, 1982). We describe this procedure in more detail in the Results section:

“In order to control for the confounding effects of Order of substance administration, we restricted our analysis on the first experimental session and re-computed the MGLM described above, leaving out predictors for Order and WMC. This procedure is recommended for crossover designs with significant carry-over effects, as the data from the first experimental session by definition are free from carry-over or task repetition effects (though at the cost of lowering the statistical power of the analyses) (Armitage and Hills, 1982).”

Moreover, we now emphasize the limitations arising from restricting our sample to the first session and performing a between-subjects comparison in the Discussion section. We clarify that this procedure lowered the power of the statistical tests, that the group difference might potentially be affected by unmeasured confounder variables, as well as that longer delays between the sessions might help to minimize the risk of task repetition effects:

“Furthermore, given possible carry-over effects in the delay of gratification task we followed the recommended approach of restricting our analysis to the data from the first session (Armitage and Hills, 1982). […] design with higher statistical power or by increasing the time between the experimental sessions to lower the risk of task repetition effects.”

2. It would be instructive to conduct some sensitivity analyses. For example the authors could tatistically control for order effects when analysing all the data in the planned within-subjects analysis to see whether the results converge. It would also be useful to report the required effect size for reasonable power (e.g. 80%) for the between-subjects analyses.

We thank the reviewers for this suggestion. First, we conducted the suggested sensitivity analysis and determined the minimum effect size that our between-subject comparison can detect with a power of 80% and an α threshold of 5%. This power analysis suggests that the between-subject comparison could detect effects sizes of Cohen’s d = 0.81 with a power of 80%. In the revised manuscript, we report this sensitivity analysis:

“A sensitivity analysis indicated that our between-subject comparisons could detect effects of Cohen’s *d* = 0.81 with a power of 80% (α = 5%).”

Regarding the reviewers’ suggestion to statistically control for order effects, we note that in all current analyses we already model the interactions of all predictors with the factor drug order, as recommended for crossover designs (Armitage and Hills, 1982). We are not aware of any further possibility to statistically control for order effects, but we would be happy to follow the reviewers’ suggestions for additional control analyses. In the revised manuscript, we now state more explicitly that including the factor Order serves to detect and control for task repetition/carry-over effects in crossover designs:

“All MGLMs included dummy-coded predictors for Substance (0 = placebo, 1 = oxytocin), WMC (-1 = low, 1 = high), and Order of substance administration (-1 = placebo in session 1 and oxytocin in session 2, 1 = oxytocin in session 1 and placebo in session 2). Substance × WMC interactions modelled potential baseline-dependent effects of oxytocin, whereas Substance × Order interactions allowed to statistically detect and control for potential order or carry-over effects of drug administration.”

3. The causal interpretation implied in the title and text, for example, "our results suggest a causal link between oxytocin and delay of gratification" (p. 11, paragraph 3), are difficult to justify based on the data reported (between-groups analysis and lack of data that prove comparability between groups at a baseline). Any casual reader only glancing at the title, abstract, and Discussion section only would assume that oxytocin increased delay of gratification within subjects, as the between-subjects analysis is not made clear. The authors should revise the title, abstract, and Discussion section to to clearly distinguish between the findings that allow for a clear causal interpretation (e.g., the impact of oxytocin on reversal learning when controlled for baseline WMC) versus those that might be confounded by pre-existing differences (e.g., the session 1 temporal discounting differences).

We thank the reviewers for this helpful comment, and we agree that the between-subject comparisons do not allow drawing causal inferences regarding the impact of oxytocin on delay discounting. Accordingly, we changed the title of the paper to: “The role of oxytocin in delay of gratification and flexibility in non-social decision making”

Also in the abstract, we now clarify that the conclusions for the delay discounting task are based on between-subject rather than on within-subject comparisons:

“In intertemporal choice, patience was higher under oxytocin than under placebo, though this difference was evident only when restricting the analysis to the first experimental session (between-group comparison) due to carry-over effects.”

Furthermore, we removed all statements that might be interpreted as suggesting a causal role of oxytocin in delay discounting throughout the entire manuscript. Finally, we now explicitly state in the limitation section that due to the group comparison one should be cautious with ascribing oxytocin a causal role for delay discounting:

“In any case, we note that one should be cautious with ascribing oxytocin a causal role for delaying gratification due to the nature of the performed between-subject comparisons.”

4. Is it correct that only WMC was assessed prior to and following drug administration? Were future orientation, restlessness, and mood were also measured before drug administration? If no, it is not appropriate to report these variables as reflecting baseline characteristics [p. 5, paragraph 2]. If yes, effects of oxytocin differences on mood, restlessness and future orientatiion should be reported (whether significant or otherwise). While there is little evidence that a single intranasal oxytocin dose has an effect on mood restlessness, it would be important to rule out such effects.

We apologize if this was not sufficiently clear in the previous manuscript version. Indeed, only WMC was assessed both prior to and after drug administration, while mood, restlessness, and future orientation were measured only after drug administration. We now clarify that only WMC represents a measure of individual baseline differences:

“We note that there was no evidence for baseline differences in WMC between oxytocin and placebo groups, *t*(46) = 0.827, *p* = 0.412, such that the oxytocin effects on delay discounting are unlikely to be explained by pre-existing group differences in baseline cognitive performance.”

For future orientation, mood, and restlessness we now report the tests for drug effects on both a within-subject and a between-subject level in order to consider the possibility that the drug effects on decision making might be driven by potential influences of oxytocin on these variables:

“Lastly, there was no evidence for oxytocin effects on future orientation, mood, or restlessness neither on a within-subject level (across both experimental sessions; future orientation: *t*(48) = 0.78, *p* = 0.437; mood: *t*(48) = 0.504, *p* = 0.617; restlessness: *t*(48) = 0, *p* = 1) or a between-subject level (restricting our analysis to the first experimental session; future orientation: *t*(46) = 0.39, *p* = 0.697; mood: *t*(46) = 0.34, *p* = 0.735; restlessness: *t*(46) = 0.157, *p* = 0.857). It is thus unlikely that the observed oxytocin effects on decision making were driven by effects on these measures. “

5. The justification for using a one-tailed p-value in the analysis of inequity preference is weak: the p value of.026 would not be significant two-tailed, and the fact that previous studies have pointed in the same direction is not a sufficiently strong justification. The main effects of inequity amount and inequity type on the outcome of the dictator game seem many times stronger than the weak and non-significant modulating effect of oxytocin.

We thank the reviewers for this comment which helped us to elaborate the justification for the one-tailed test in the dictator game. According to Ruxton and Neuhäuser (2010), the use of one-tailed tests is justified if authors are more interested in an effect into one direction than another, and this is the case here because we wanted to test whether or not we can replicate previous findings by Pornpattananangkul, Zhang, Chen, Kok, and Yu (2017). Given these previous findings, even an effect in the unexpected direction would have been considered as a failed replication. This is why we had pre-registered a directional hypothesis for the impact of oxytocin on advantageous inequity aversion (https://osf.io/ykvd5). In the revised manuscript, we elaborated the justification for using a one-tailed test:

“The α threshold was set to 5% two-tailed for all analyses except for the dictator game where we used a one-tailed test (as pre-registered) to replicate previous findings that oxytocin increases generosity (Pornpattananangkul et al., 2017). Given these previous findings, we would consider both a too-weak effect in the expected direction and an effect in the unexpected direction as failed replication of previous reports that oxytocin increases advantageous inequity aversion.”

Finally, we now explicitly mention in the revised manuscript that the oxytocin effects are weaker than the influences of task-specific manipulations (e.g., degree of inequity) and that the oxytocin effect on advantageous inequity would be only marginally significant with a two-tailed test:

“We note, though, that the impact of oxytocin on advantageous inequity aversion was weaker than the effects of other task-specific experimental manipulations (e.g., Inequity amount) and would have been only marginally significant (*p* = 0.052) with a two-tailed test.”

6. The analyses contain a large number of main effects and interactions. For full transparency it is important to report these for example by including tables showing the complete models (this could be included in a supplement). It should be explicitly stated whether any effects other than those reported reached significance, for example using wording such as "no other factors or interactions reached significance, all p>.05."

We thank the reviewer for the suggestion. We created a Supplementary materials file where we report the full results of our statistical models (Tables S1-S7). Moreover, in the main text we now report all significant effects and interactions and clarify that no further effect reached significance.

7. There needs to be some discussion of the fact that oxytocin induced a practice effect in delay discounting but not the other tasks. For example, reversal learning often results in practice effects, yet, in this study these did not appear to interact with oxytocin. Do the authors consider this to be meaningful, and what could account for such differences across tasks? How could this set of results inform future research?

We agree that it is worth discussing why only the intertemporal choice task showed strong carry-over effects. As we did not expect apriori to observe such task-specific carry-over effects, we can only speculate regarding the reasons for this. One possibility is that anchoring effects are stronger in the intertemporal choice task than in the reversal learning task, i.e. participants might have remembered their choices in the first experimental session and have tried to make consistent choices in the second session. In other words, participants who had received oxytocin already in the first experimental session might have made more patient choices also in the second session under placebo because they remembered their relatively more patient choices under oxytocin in the first session. Consistent with this, participants generally made more patient choices in the second (mean = 73.4%) than in the first session (mean = 70.6%). We discuss this potential reason for the carry-over effects in the revised manuscript:

“We can only speculate about the reasons for the repetition effects in the intertemporal choice task. The fact that choices were more patient in the second (mean = 73.4%) than in the first experimental session (70.6%) indicates potential anchoring effects, such that participants receiving placebo in the second session might have remembered their relatively more patient choices under oxytocin in the first session.“

8. The Discussion section would be strengthened if there was a greater focus on discussing the results that were observed. There is quite a lot of speculation (e.g., "by strengthening valence-unspecific prediction error signals in the striatum…") that is rarely labelled as such. The speculation needs to be toned down considerably, and where necessary it should be flagged with language such as "We speculate that…".

In the revised Discussion section, we now clearly mark speculations as such:

“We speculate that oxytocin might have reduced impulsiveness via interactions with the dopaminergic system, as delay of gratification has been related to dopaminergic activity (Pine, Shiner, Seymour, and Dolan, 2010; Weber et al., 2016), whereby blocking dopaminergic neurotransmission increases delay of gratification similar to our current findings.”

“We therefore speculate that oxytocin might have improved reversal learning by strengthening valence-unspecific prediction error signals in the striatum, in analogy to previous findings for the dopamine antagonist sulpiride in a combined pharmacology-neuroimaging study (van der Schaaf et al., 2014).”

“It is thus tempting to speculate that oxytocin might have influenced decision making via oxytocin receptors expressed in striatal reward circuits, where oxytocin might mirror the effects of dopamine antagonists (Love, 2014).”

“According to this view, a plausible interpretation of our findings might be that oxytocin could have increased prosocial giving and delay of gratification by lowering the subjective values of selfish and immediate rewards, respectively, whereas it may have facilitated reversal learning by enhancing neural prediction error signals.”

References:

Armitage, P., and Hills, M. (1982). The two‐period crossover trial. *Journal of the Royal Statistical Society: Series D, 31*(2), 119-131.

Pornpattananangkul, N., Zhang, J., Chen, Q., Kok, B. C., and Yu, R. (2017). Generous to whom? The influence of oxytocin on social discounting. *Psychoneuroendocrinology, 79*, 93-97. doi:10.1016/j.psyneuen.2017.02.016

Ruxton, G. D., and Neuhäuser, M. (2010). When should we use one‐tailed hypothesis testing? *Methods in Ecology Evolution, 1*(2), 114-117.